# Rethinking Negative Pairs in Code Search

**Haochen Li**[1]  **Xin Zhou**[2]  **Luu Anh Tuan**[1]  **Chunyan Miao**[1]

[1]School of Computer Science and Engineering, Nanyang Technological University, Singapore
[2]Alibaba-NTU Singapore Joint Research Institute, Nanyang Technological University, Singapore
`{haochen003, xin.zhou, anhtuan.luu, ascymiao}@ntu.edu.sg`

## Abstract

Recently, contrastive learning has become a key component in fine-tuning code search models for software development efficiency and effectiveness. It pulls together positive code snippets while pushing negative samples away given search queries. Among contrastive learning, InfoNCE is the most widely used loss function due to its better performance. However, the following problems in negative samples of InfoNCE may deteriorate its representation learning: 1) The existence of false negative samples in large code corpora due to duplications. 2). The failure to explicitly differentiate between the potential relevance of negative samples. As an example, a bubble sorting algorithm example is less "negative" than a file saving function for the quick sorting algorithm query. In this paper, we tackle the above problems by proposing a simple yet effective Soft-InfoNCE loss that inserts weight terms into InfoNCE. In our proposed loss function, we apply three methods to estimate the weights of negative pairs and show that the vanilla InfoNCE loss is a special case of Soft-InfoNCE. Theoretically, we analyze the effects of Soft-InfoNCE on controlling the distribution of learnt code representations and on deducing a more precise mutual information estimation. We furthermore discuss the superiority of proposed loss functions with other design alternatives. Extensive experiments demonstrate the effectiveness of Soft-InfoNCE and weights estimation methods under state-of-the-art code search models on a large-scale public dataset consisting of six programming languages. Source code is available at `https://github.com/Alex-HaochenLi/Soft-InfoNCE`.

## 1 Introduction

Code search is a common activity in software development that can boost the productivity of software developers (Nie et al., 2016; Shuai et al., 2020). Code search models can retrieve code fragments relevant to a given query from code bases (Grazia

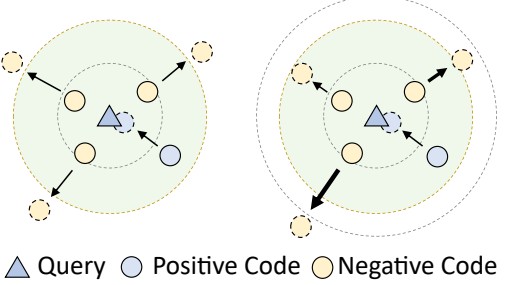

△ Query  ◯ Positive Code  ◯ Negative Code

Figure 1: Contrastive learning pushes away negative pairs in the representation space. **Left:** Existing works treat negative pairs equally. **Right:** Negative pairs should be pushed away according to their similarity with the query. A thicker arrow means that this sample is more negative than others.

and Pradel, 2022). To train or fine-tune code search models, contrastive learning has become a key component in learning discriminative representations of queries and codes as it pushes apart negative query-code pairs and pulls together positive pairs (Shi et al., 2022a; Li et al., 2022a). InfoNCE (Van den Oord et al., 2018) is a representative choice of contrastive learning loss that considers the other in-batch samples as negative pairs given a query (Huang et al., 2021).[1]

Although InfoNCE is effective in code search, we argue it is sub-optimal at discriminating code samples as it suffers from the following problems. First, the existence of false negatives in code bases. Lopes et al. (2017); Allamanis (2019) finds that code duplications are common in large code corpus, which means that many negative pairs are in fact false negatives. Training with false negative pairs may deteriorate code representation learning. Second, the setting of InfoNCE ignores the potential relevance of negative codes (Li et al., 2022c). For example, for a given query asking about quick sort-

---

[1]We conduct a survey to demonstrate the dominant adoption of InfoNCE. See Appendix A for details.

ing algorithms, among negative codes the bubble sorting algorithm is expected to be retrieved before file-saving functions, but the current training procedure cannot model this relationship explicitly. The InfoNCE loss just treats all negative codes equally, as shown in Fig.1. In fact, the false negative cancellation is a special case of modeling potential relevance, where the former only considers potential relevance as binary while the latter focus on describing it in a continuous field. Although some methods (Huynh et al., 2022; Chen et al., 2022; Li et al., 2022c) are proposed to solve these two problems, they are all applied during model pre-training. In model fine-tuning stage, they still use InfoNCE loss. As a result, our aforementioned two problems are still largely unexplored in model fine-tuning stage.

In this work, we first revisit the commonly used InfoNCE loss and explain why it cannot model potential relations among codes explicitly. Then we present Soft-InfoNCE to handle this problem, by simply inserting a weight term into the denominator of InfoNCE loss. We also propose three methods to estimate the weight terms and compare them empirically. To justify the effect of Soft-InfoNCE loss, we theoretically analyze its properties with regard to representation distribution and mutual information estimation. Our analysis indicates that the inserted weight encourages negative pairs to approximate a given distribution and reduces bias in the estimation of mutual information by leveraging importance sampling. Moreover, we prove that the proposed Soft-InfoNCE loss upper bounds other loss function designs like Binary Cross Entropy and weighted InfoNCE loss. We also relate existing false negative cancellation methods and our methods. Finally, we demonstrate the effectiveness of the proposed Soft-InfoNCE loss by evaluating it on several pre-training models across six large-scale datasets. Additional ablation studies also validate our theoretical analysis empirically.

In summary, our contributions of this work are as follows:

- We propose a novel contrastive loss, Soft-InfoNCE, that models the potential relations among negative pairs explicitly by simply inserting a weight term to the vanilla InfoNCE loss.

- We conduct theoretical analysis to show that Soft-InfoNCE loss can control the distribution of learnt representations and reduce the

bias in mutual information estimation. We also prove the superiority of Soft-InfoNCE loss over other choices of design and reach a conclusion that previous false negative cancellation works can be considered as a special case of our proposed methods by discussing the relation.

- We apply Soft-InfoNCE loss on several code search models and evaluate them on the public CodeSearchnet dataset with six programming languages. Extensive experiment results verify the validity of our theoretical analysis and the effectiveness of our method.

## 2 Preliminaries

Code search aims at retrieving the most relevant code fragments for a given query. During training, we take the comment of code as a query and maximize the similarities between the query and its associated code. Meanwhile, we minimize the similarities between negative pairs that are generated by In-Batch Negative (Huang et al., 2021) strategy. Given a code distribution $\mathbf{C} = \{x_i\}_{i=1}^K$ and a comment distribution $\mathbf{Q} = \{y_i\}_{i=1}^K$, where $x_i$ is a code fragment and $y_i$ is its corresponding query, $K$ is the size of the dataset, a Siamese encoder $g : \mathbf{C} \cup \mathbf{Q} \to \mathbf{H}$ is used to map codes and queries to a shared representation space $\mathbf{H}$. Thus, we obtain two representation sets, $\mathbf{H_c} = \{g(x_i)\}_{i=1}^K$ and $\mathbf{H_q} = \{g(y_i)\}_{i=1}^K$.

We calculate the similarities between query-code pairs by dot product or cosine distance. And we optimize the distribution of representations by contrastive learning. Several loss functions are proposed for this objective (Mikolov et al., 2013; Weinberger and Saul, 2009; Hadsell et al., 2006). Among them, InfoNCE loss (Van den Oord et al., 2018) is dominantly adopted by recent code search models due to its better performance than others. We denote $q_i \in \mathbf{H_q}$ and $c_i \in \mathbf{H_c}$ as representations of queries and codes, respectively. For a given batch of data, we could generate 1 positive pair and $N - 1$ negative pairs for each query, where $N$ is the batch size. The InfoNCE loss can be described as:

$$\mathcal{L} = -\mathbb{E}\left[\log \frac{\exp(q_i \cdot c_i)}{\exp(q_i \cdot c_i) + \sum_{j \neq i}^N \exp(q_i \cdot c_j)}\right] \tag{1}$$

where $(q_i, c_j)$ are positive pairs when $i = j$ and negative pairs otherwise. Here we adopt the dot product as the measurement of similarity.

## 3 Our Method

### 3.1 Revisiting InfoNCE Loss

We reformulate Eq.(1) to:

$$\mathcal{L} = -\mathbb{E}\left[q_i \cdot c_i\right]$$
$$+ \mathbb{E}\left[\log\left(\exp(q_i \cdot c_i) + \sum_{j \neq i}^{N} \exp(q_i \cdot c_j)\right)\right]. \tag{2}$$

As discovered by (Wang and Isola, 2020), the two terms correspond to two objectives of contrastive learning. The first term can be expressed as the alignment of positive pairs that pulls positive instances together. The second term enforces uniform distribution of negative pairs because it pushes all negative pairs apart.

However, we argue that negative pairs should not be distributed uniformly. In other words, unlabeled data may also share some similarities with the given query. Suppose we have a query "How to implement a bubble sorting algorithm in Python?", although both a quick sorting algorithm and a file saving function are considered to be negative results, quick sorting is expected to be retrieved before file saving since it is more relevant to the query. Moreover, Lopes et al. (2017); Allamanis (2019) find that code duplication is common in code corpora which means that there are many false negative examples during training. In conclusion, negative samples in a batch should not be treated equally, as illustrated in Fig.1.

### 3.2 Soft-InfoNCE Loss

To address the aforementioned problems, we propose Soft-InfoNCE loss by simply inserting a weight term $w_{ij}$ into the original format, which can be described as:

$$\mathcal{L} = -\frac{1}{N}\sum_{i=1}^{N}\left[\log \frac{\exp(q_i \cdot c_i)}{\exp(q_i \cdot c_i) + \sum_{j \neq i}^{N} w_{ij}\exp(q_i \cdot c_j)}\right], \tag{3}$$

$$w_{ij} = \frac{\beta - \alpha \cdot sim_{ij}}{\beta - \frac{\alpha}{N-1}\sum_{j \neq i}^{N} sim_{ij}}, \tag{4}$$

where $sim_{ij} \in [0, 1]$, $\sum_{j \neq i}^{N} sim_{ij} = 1$, and $\alpha, \beta$ are hyper-parameters. $sim_{ij}$ is the similarity score

between query $q_i$ and code $c_j$. The numerator in Eq.(4) puts smaller weights on less negative codes, and the denominator is a normalization factor to make sure $\sum_{j \neq i}^{N} w_{ij} = N - 1$, which is the same value as in the vanilla InfoNCE loss. Now we can derive that the gradients of negative pairs are proportionally related to $w_{ij}$:

$$\frac{\partial \mathcal{L}}{\partial \exp(q_i \cdot c_j)} = \frac{w_{ij}}{N \sum_{j=1}^{N} w_{ij}\exp(q_i \cdot c_j)}. \tag{5}$$

Note that $w_{ij} = 1$ in this equation when $i = j$. The vanilla InfoNCE loss is a special case of Eq.(3), which sets all $w_{ij}$ as 1. Models may learn implicit relationships between a query and different negative samples under the vanilla InfoNCE loss, but we argue that modeling this relationship explicitly by $w_{ij}$ has a positive influence on learning better representations.

Then comes the estimation of similarity score $sim_{ij}$. The ideal solution is using human-annotated labels. However, there are no existing datasets and it is challenging to label all the possible negative pairs since the number of them increases quadratically with the number of positive pairs (e.g., 100 positive pairs can generate 9900 negative pairs). Thus, we employ the following approaches to estimate the similarity scores $sim_{ij}$ and empirically analyze and compare them in Section 6.

**BM25.** BM25 is an enhanced version of TF-IDF, which matches certain terms in codes with the given query.

**SimCSE.** Unsupervised SimCSE (Gao et al., 2021) is a recently proposed method that has outstanding performance on sentence similarity tasks. We measure the similarity between a query and unlabeled code indirectly by measuring that between the query and code's positive query. The assumption behind this is that a query and its positive code are exactly matched so that they are perfectly aligned in the representation space.

**Trained Model.** Models that are trained with vanilla InfoNCE loss on datasets may have certain capabilities to correctly predict the similarity.

Note that for the last two estimation methods, we load and freeze their pre-trained parameters during training. After calculating the similarity scores, we normalize the results with Softmax function with temperature $t$ to satisfy $\sum_{j \neq i}^{N} sim_{ij} = 1$.

## 4  Justification

In this section, we analyze the properties of Soft-InfoNCE loss and compare it with related works to justify its effectiveness. Recall that the optimization objective of vanilla InfoNCE loss can be divided into two parts, alignment and uniformity. The insertion of $w_{ij}$ only has an influence on the second term. Thus, for simplicity, we analyze the second term in this section, which is:

$$\mathcal{L}_{unif} = \frac{1}{N}\sum_{i=1}^{N}\Big[\log\Big(\exp(q_i \cdot c_i)+ \\ \sum_{j\neq i}^{N} w_{ij}\cdot\exp(q_i\cdot c_j)\Big)\Big]. \quad (6)$$

### 4.1  Effect on Representation Distribution

Intuitively, we can control the distribution of negative samples by setting different weights. Here we theoretically prove that Soft-InfoNCE loss upper bounds the KL divergence between the predicted similarity of negative pairs and $sim_{ij}$.

**Theorem 1** *For a batch of query representations $\{q_i\}_{i=1}^{N}$, code representations $\{c_i\}_{i=1}^{N}$, and similarity scores $\mathbf{S_i} = \{sim_j\}_{j\neq i}^{N}$, we have:*

$$\mathcal{L}_{unif} \geq \frac{1}{N(\beta N - \alpha - 1)}\sum_{i=1}^{N}\Big[\beta\sum_{j\neq i}^{N}\log P_\theta(c_j|q_i) \\ + \alpha KL(\mathbf{S_i}|P_\theta(c_j|q_i))\Big] + const.w.r.t.\ \mathbf{S_i}., \quad (7)$$

*where $N$ is batch size, $P_\theta(c_j|q_i)$ is predicted similarity between query $q_i$ and code $c_j$ by model $\theta$.*

Proof of Theorem 1 is presented in Appendix B.1. We can observe that in addition to the original objective which minimizes the predicted similarity scores of negative pairs, the second term encourages the similarity distribution to fit the given distribution $\mathbf{S_i}$. When we set $\alpha = \beta = 1$ and $sim_{ij} = \frac{1}{N-1}$, it becomes the vanilla InfoNCE loss hence all $w_{ij} = 1$. This could also be used as an explanation for the uniformity objective of vanilla InfoNCE loss.

### 4.2  Effect on Mutual Information Estimation

It has already been proved that optimizing InfoNCE loss improves the lower bounds of mutual information for a positive pair (Van den Oord et al., 2018).

Since the optimal value for $\exp(q\cdot c)$ is given by $\frac{p(c|q)}{p(c)}$, the derivation can be described as follows:

$$\mathcal{L} = -\mathbb{E}\log\left[\frac{\frac{p(c_i|q_i)}{p(c_i)}}{\frac{p(c_i|q_i)}{p(c_i)} + \sum_{j\neq i}^{N}\frac{p(c_j|q_i)}{p(c_j)}}\right]$$

$$= \mathbb{E}\log\left[1 + \frac{p(c_i)}{p(c_i|q_i)}\sum_{j\neq i}^{N}\frac{p(c_j|q_i)}{p(c_j)}\right] \quad (8)$$

$$\approx \mathbb{E}\log\left[1 + \frac{p(c_i)}{p(c_i|q_i)}(N-1)\underset{c_j\in\mathbf{C_{neg}}}{\mathbb{E}}\frac{p(c_j|q_i)}{p(c_j)}\right] \quad (9)$$

$$\geq \mathbb{E}\log\left[\frac{p(c_i)}{p(c_i|q_i)}N\right]$$

$$= -I(q_i,c_i) + \log N,$$

where $\mathbf{C_{neg}}$ is the whole set of negative codes for the given query $q_i$. A key step in the above derivation is the approximation from Eq.(8) to Eq.(9). In Eq.(8) we calculate the sum of $\frac{p(c_j|q_i)}{p(c_j)}$ in a batch to estimate that of the whole negative set. As Van den Oord et al. (2018) mentioned, Eq.(8) becomes more accurate when $N$ increases. By adding a constant term $\log(N-1)$, it would be clearer that InfoNCE loss builds a Monte-Carlo estimation sampling from a uniform distribution:

$$\mathcal{L} + \log(N-1)$$

$$= -\mathbb{E}\log\left[\frac{\frac{p(c_i|q_i)}{p(c_i)}}{\frac{1}{N-1}\frac{p(c_i|q_i)}{p(c_i)} + \frac{1}{N-1}\sum_{j\neq i}^{N}\frac{p(c_j|q_i)}{p(c_j)}}\right] \quad (10)$$

Incorporating $w_{ij}$ in Eq.(10) reduces estimation bias since it can be considered as using an importance sampling strategy, resulting in more precise estimation. Specifically, we want to estimate the expectation of $\frac{p(c_j|q_i)}{p(c_j)}$ according to a real distribution between the query $q_i$ and negative codes $c_j$ in the context of code search. However, the negative codes in a batch are randomly sampled which follows a uniform distribution. To bridge this gap and get an unbiased estimator, importance sampling is adopted by inserting a weight term $w_{ij}$. We denote the uniform distribution as $p$ and the real distribution as $q$. Thus, calculating expectations based on $q$ could be derived from $p$:

$$\mathbf{E_{c_j\sim q}}\left[\frac{p(c_j|q_i)}{p(c_j)}\right] = \mathbf{E_{c_j\sim p}}\left[\frac{p(c_j|q_i)}{p(c_j)}\cdot\frac{q(c_j)}{p(c_j)}\right]$$

$$\mathbf{E}_{\mathbf{c_j}\sim\mathbf{q}}\left[\frac{p(c_j|q_i)}{p(c_j)}\right] = \frac{1}{N-1}\sum_{j\neq i}^{N}\frac{p(c_j|q_i)}{p(c_j)}$$

If we take both $\alpha$ and $\beta$ as 1 in Eq.(4), we have:

$$\mathbf{E}_{\mathbf{c_j}\sim\mathbf{p}}\left[\frac{p(c_j|q_i)}{p(c_j)}\cdot\frac{q(c_j)}{p(c_j)}\right]$$
$$= \frac{(N-1)(1-sim_{ij})}{N-2}\sum_{j\neq i}^{N}\frac{p(c_j|q_i)}{p(c_j)} \qquad (11)$$

And we could find that $q(c_j) = \frac{1-sim_{ij}}{N-2}$, which is inversely proportional to $sim_{ij}$. This property is in line with our intuition that we should consider true negatives more when approximating the whole negative set. As we know, importance sampling provides more accurate estimates when the sampling distribution is closer to the real distribution. We hypothesize that for a given query, the majority of codes share a low similarity, which makes it suitable to use the Softmax function to normalize $sim_{ij}$. We will empirically discuss this hypothesis in Section 6.

### 4.3 Relation with Other Loss Functions

In this part, we connect Soft-InfoNCE loss with other loss functions by analyzing them theoretically. Besides, in Section 6 we report their comparative evaluation results empirically.

**Binary Cross Entropy Loss.** We may consider $sim_{ij}$ as soft labels and hence use Binary Cross-Entropy (BCE) loss to train the model. The probability in BCE is calculated by $\frac{\exp(q_i\cdot c_i)}{\sum_{j=1}^{N}\exp(q_i\cdot c_j)}$. While Soft-InfoNCE loss upper bounds the KL divergence, BCE lower bounds it, which is described in Theorem 2.

**Theorem 2** *For a batch of query representations $\{q_i\}_{i=1}^{N}$, code representations $\{c_i\}_{i=1}^{N}$, and similarity scores $\mathbf{S_i} = \{sim_j\}_{j\neq i}^{N}$, we have:*

$$\mathcal{L}_{BCE} \leq -\frac{1}{N^2}\sum_{i=1}^{N}\left[\sum_{j\neq i}^{N}\log(1-P_\theta(c_j|q_i))\right.$$
$$\left. + KL(\mathbf{S_i}|P_\theta(c_j|q_i)) + const.w.r.t.\ \mathbf{S_i}\right], \qquad (12)$$

*where $N$ is batch size, $P_\theta(c_j|q_i)$ is predicted similarity between query $q_i$ and code $c_j$ by model $\theta$.*

Proof of Theorem 2 is presented in Appendix B.2. In general, minimizing an upper bound leads to better performance compared with lower bounds.

**Weighted InfoNCE Loss.** Another choice is using weighted InfoNCE loss. We take $sim_{ij}$ as weights, which can be described as:

$$\mathcal{L}_W = -\mathbb{E}\left[sim_{ij}\cdot\log\frac{\exp(q_i\cdot c_i)}{\sum_{j=1}^{N}\exp(q_i\cdot c_j)}\right]. \qquad (13)$$

Note that we set the weight $sim_{ii}$ for the positive pair as 1.

**Proposition 1** *Soft-InfoNCE loss upper bounds weighted InfoNCE loss:*

$$\mathcal{L}_{Soft} \geq \mathcal{L}_W + \log P_\theta(c_i|q_i)$$
$$+ \sum_{i\neq j}\frac{N-1-sim_{ij}}{N-2}\log P_\theta(c_j|q_i). \qquad (14)$$

*The equilibrium satisfies when all code fragments in the batch are actually false negatives.*

The proof of Proposition 1 is presented in Appendix B.3. Without loss of generalization, our proof is deduced based on one query, however, it also holds for a batch of queries. From the above proposition, we can find that Soft-InfoNCE loss $\mathcal{L}_{Soft}$ upper bounds Weighted InfoNCE loss $\mathcal{L}_W$. Thus, one would expect $L_{Soft}$ to be the superior loss function.

**KL Divergence Regularization.** As proved in Theorem 1, a KL divergence term that measures the similarity between the given distribution $\mathbf{S}$ and predicted distribution $P_\theta$ is incorporated during the optimization implicitly. We try to explicitly add a KL divergence regularization term to the vanilla InfoNCE loss and compare its performance with our proposed loss empirically.

### 4.4 Relation with False Negative Cancellation

Recently, several works focus on eliminating the effect of false negative samples by first detecting those samples and then removing them from the negative sets. Though these detection methods are different among tasks, their cancellation operations share the same principle. That is, false negatives are removed from the denominator of the InfoNCE loss. It can be considered as a special case of our proposed Soft-InfoNCE loss, which sets the weights $w_{ij}$ of false negatives as 0 while others remain 1. False negative cancellation methods are effective in classification or unsupervised

pre-training tasks since they only need to consider whether negative samples belong to the same class or not. However, in the context of a similarity-based retrieval task, models are required to discriminate negative samples by continuous values. In Section 6, we report an empirical comparison between cancellation methods and ours.

## 5 Experimental Setup

In this section, we elaborate on the evaluated dataset, baselines, and our implementation details.

**Datasets.** We use a large-scale benchmark dataset **CodeSearchNet** (CSN) (Husain et al., 2019) to evaluate the effectiveness of Soft-InfoNCE loss which contains six programming languages including Ruby, Python, Java, Javascript, PHP, and Go. The dataset is widely used in previous studies (Feng et al., 2020; Guo et al., 2021, 2022) and the statistics are shown in Appendix C.1. For the training set, it contains positive-only query-code pairs while for the validation and test sets the model attempts to retrieve true code fragments from a fixed codebase. We follow (Guo et al., 2021) to filter out low-quality examples. The performance is measured by the widely adopted the Mean Reciprocal Rank (MRR) which is the average of reciprocal ranks of the true code fragment for a given query. It can be calculated as:

$$MRR = \frac{1}{|Q|} \sum_{i=1}^{|Q|} \frac{1}{Rank_i}, \qquad (15)$$

where $Rank_i$ is the rank of the true code fragment for the $i$-th given query $Q$.

**Baselines.** We apply Soft-InfoNCE loss on several code search models: **CodeBERT** is a bi-modal pre-trained model pre-trained on mask language modeling and replaced token detection (Feng et al., 2020). Note that in this work we refer CodeBERT to the siamese network architecture described in the original paper. **GraphCodeBERT** incorporates the structure information of codes and further develops two structure-based pre-training tasks: node alignment and data flow edge prediction (Guo et al., 2021). **UniXCoder** unifies understanding and generation pre-training tasks to enhance code representation leveraging cross-model contents like Abstract Syntax Trees (Guo et al., 2022).

**Implementation Details.** For all the settings related to model architectures, we follow the original paper. For hyper-parameter settings that affect the

calculation of Soft-InfoNCE loss, we provide implementation details in Appendix C.2. For BM25 estimation, we merely measure similarity based on in-batch data. For trained model estimation, we train the same model of each studied model following the training settings of the original paper on different programming languages separately until convergence. For SimCSE estimation, we first initialize the model with the HuggingFace released parameters[2] and then train it following the default setting of the original paper. Note that we collect all the natural language queries from different programming language training sets to boost performance on SimCSE unsupervised learning. The training epoch is set to 30 for all studied models. Experiments described in this paper are running with 3 random seeds 1234, 12345, and 123456. All experiments meet p<0.01 of significance tests except for the results of GraphCodeBERT and UniX-Coder on the Go dataset. Experiments are conducted on a GeForce RTX A6000 GPU.

## 6 Results

In this section, we first show the overall performance of three weight estimation approaches when applying Soft-InfoNCE loss. Then, we empirically compare our proposed method with other choices of loss functions and false negative cancellation methods. Finally, we conduct an ablation study to analyze the effect of hyper-parameters.

**Overall Results.** The results in Table 1 reveal that baseline models equipped with Soft-InfoNCE loss can gain an overall 1%-2% performance improvement in MRR over that of InfoNCE loss across six programming languages. The consistent improvements observed in all three estimation approaches demonstrate the effectiveness of our proposed Soft-InfoNCE loss in code search. Time efficiency comparison is also performed in Appendix D.1.

**Comparison among Estimation Approaches.** As shown in Table 1, trained-model estimation significantly outperforms the other two methods on CodeBERT, while all estimation approaches improve on the other two models to a similar extent. Among them, SimCSE is the most robust one concerning model types.

---

[2] https://huggingface.co/sentence-transformers/msmarco-distilbert-dot-v5

| Model | Loss | Estimator | Ruby | Python | Java | JavaScript | PHP | Go |
|---|---|---|---|---|---|---|---|---|
| CodeBERT | InfoNCE | - | 0.648 | 0.636 | 0.663 | 0.594 | 0.615 | 0.878 |
|  | Soft-InfoNCE | BM25 | 0.660 | 0.664 | **0.682** | 0.600 | 0.621 | 0.882 |
|  |  | Trained Model | **0.682** | **0.675** | **0.682** | **0.615** | **0.631** | **0.898** |
|  |  | SimCSE | 0.666 | 0.668 | 0.670 | 0.607 | 0.623 | 0.887 |
| GraphCodeBERT | InfoNCE | - | 0.705 | 0.690 | 0.691 | 0.647 | 0.648 | **0.896** |
|  | Soft-InfoNCE | BM25 | **0.730** | 0.697 | 0.698 | 0.652 | 0.655 | 0.892 |
|  |  | Trained Model | 0.719 | 0.692 | 0.692 | 0.653 | 0.648 | 0.889 |
|  |  | SimCSE | 0.721 | **0.700** | **0.702** | **0.656** | **0.657** | 0.894 |
| UniXCoder | InfoNCE | - | 0.740 | 0.720 | 0.726 | 0.684 | 0.676 | 0.915 |
|  | Soft-InfoNCE | BM25 | **0.753** | **0.728** | **0.733** | 0.693 | **0.684** | **0.916** |
|  |  | Trained Model | **0.753** | **0.728** | 0.731 | 0.694 | 0.682 | 0.915 |
|  |  | SimCSE | **0.753** | 0.726 | 0.731 | **0.699** | **0.684** | 0.913 |

Table 1: Results of different weight estimation approaches under MRR.

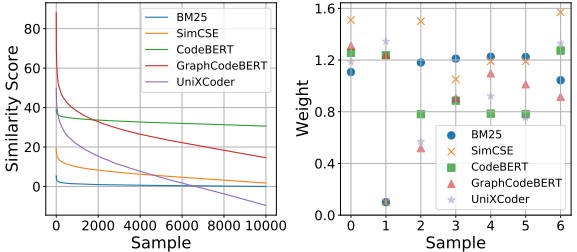

Figure 2: Similarity estimations by different approaches for a random batch of samples in CSN-Python.

In Fig.2, we take a random batch of samples from CSN-Python to analyze the difference among estimation approaches. From the left figure, we can find that predicted similarity scores roughly follow a softmax distribution, per our hypothesis. As for the right one, we can see that compared with other estimation methods, BM25 generates similar weights for the majority of negative samples. This is because BM25 calculates similarities only based on keyword matching and for most negative codes there is no keyword overlapping at all. While for neural model based methods, they can capture latent semantic similarities. Considering the estimated weights of samples from trained models in the right panel of Fig. 2, we find that some weights that are predicted by the three models are similar while others are not, or even contradictory with each other.

To better understand the differences among these estimation methods, we perform case studies for the eight samples in the right panel of Fig. 2 in Appendix D.2. From the case study, we find that

there are contradictory predictions when we use InfoNCE-tuned code search models to predict $w_{ij}$. This phenomenon indicates that although existing code search models could find true code snippets well, they cannot recognize potential relevance among negative codes, which is the core motivation for proposing Soft-InfoNCE. Kindly note that in this paper we do not investigate estimation methods fully but mainly focus on the effectiveness of Soft-InfoNCE, as discussed in the 'Limitations' Section.

**Comparison with Other Loss Functions.** Table 2 shows the overall performance of different loss functions. In Table 5, we also give the detailed results for each programming language. For the calculation of these loss functions, we follow the definition described in Sec.4.3. Note that for KL regularization we set the weights of the original loss and regularization term as 1.3 and 0.7 to fairly compare with Soft-InfoNCE loss, and we use SimCSE estimation for all experiments. We can see significant drops of MRR compared with Soft-InfoNCE loss, which is in agreement with our theoretical analysis. We think those three loss functions somewhat improve the mutual information of negative pairs. Take weighted InfoNCE as an example. Though weights for negative pairs are only at around 0.03, optimizing negative pairs still lower bounds the mutual information with a constant $\log N$ according to Eqn.9. It makes feature vectors distribute closer and hence hard to distinguish each other, which makes the performance even worse than InfoNCE. The analysis also works for BCE and KL regularization, which try to im-

| Loss | CodeBERT | GraphCodeBERT | UniXCoder |
|------|----------|---------------|-----------|
| BCE | 0.641 | 0.679 | 0.686 |
| W-InfoNCE | 0.639 | 0.674 | 0.714 |
| KL Reg. | 0.648 | 0.688 | 0.716 |
| Ours | **0.687** | **0.722** | **0.751** |

Table 2: Overall performance of different loss function designs on six programming languages under MRR. "W-InfoNCE" stands for weighted InfoNCE loss and "KL Reg." stands for KL regularization.

| Methods | | MRR | Methods | | MRR |
|---------|---|------|---------|------|------|
| | 1 | 0.685 | | 0.7 | 0.689 |
| Top-K | 3 | 0.681 | Dynamic Threshold | 0.5 | 0.688 |
| | 5 | 0.680 | | 0.3 | 0.688 |

Table 3: Performance of false negative cancellation methods applied to GraphCodeBERT on CSN-Python.

prove the similarity between a query and negative codes as well. Therefore, we argue negative pairs should not be placed at the numerator of InfoNCE.

**Comparison with False Negative Cancellation Methods.** We apply two types of false negative cancellation methods and evaluate their performance, as shown in Table 3. The first type involves removing top-K similar negatives. We observe that performance decreases when more negative samples are removed because sometimes there are no false negative samples in the batch. Hence, removing top-k negatives directly may accidentally remove hard negative samples. The other option is the dynamic threshold. It takes negative samples that have similarities greater than certain ratios of the positive sample as false negatives. Besides having the same drawbacks as the top-k method, it is also hard to determine an appropriate ratio. Thus, there are slight drops when applying the dynamic threshold. While using Soft-InfoNCE can achieve a MRR of 0.700 which is better than the result in Table 3, we argue that setting a weight on negatives would be less risky than removing them directly.

**Effect of $\alpha$ and $\beta$.** $\alpha$ and $\beta$ control the weights of two terms in Theorem 1, one for minimizing predicted similarities and the other for KL divergence, which contradicts each other to some extent. To balance training, we performed empirical experiments to guide the setting of these two hyper-parameters, as shown in Fig.3. Note that we follow $\frac{\alpha+\beta}{2} = 1$ to make Soft-InfoNCE fall on the same scale as the

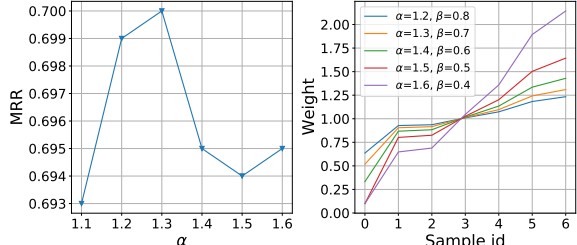

Figure 3: MRR and estimated weights under different $\alpha$ and $\beta$ settings on GraphCodeBERT over CSN-Python.

original InfoNCE loss. The left part of Fig.3 shows that the performance of Soft-InfoNCE is relatively stable with different settings and $\alpha = 1.3, \beta = 0.7$ reaches its best performance. Besides, since $\alpha$ and $\beta$ are incorporated into the calculation of weights, they also have effects on weights estimation. As shown in the right part of Fig.3, the increasing of $\alpha$ makes the weights more distinct.

## 7 Related Works

**Code Search Models.** There are mainly three stages in the development of code search models. Traditional information retrieval techniques match keywords between queries and code fragments (Hill et al., 2011; Yang and Huang, 2017; Satter and Sakib, 2016; Lv et al., 2015; Van Nguyen et al., 2017). Since natural language and programming language have different syntax rules, they often suffer from vocabulary mismatch problems (McMillan et al., 2011). Then, with the popularity of neural networks, several methods are proposed to better capture the semantics of both queries and codes (Gu et al., 2021; Cambronero et al., 2019; Gu et al., 2018; Husain et al., 2019). Generally, they are encoded by neural encoders into a shared representation space. Recently, transformer-based pre-trained models significantly outperformed previous methods. CodeBERT (Feng et al., 2020) is pre-trained via masked language modeling and replaced token detection. GraphCodeBERT (Guo et al., 2021) leverages data flow as additional information to model the relationship among variables. UniXCoder (Guo et al., 2022) is a model that can support understanding and generation tasks at the same time. This allows it to further boost performance by using the pre-training tasks (e.g. unidirectional language modeling, denoising objective) both. In this work, we mainly consider pre-trained models due to their better performance.

**False Negatives in Unsupervised Contrastive**

**Learning.** The false negative problem in unsupervised contrastive learning has been studied by some researchers. Since in contrastive learning we automatically consider other in-batch examples as negatives, we may sample false negatives during training, which results in discarding semantic information and slow convergence. Several approaches are proposed to detect those false negatives. The key insight behind these detection methods is that false negative examples are similar to positive ones. Huynh et al. (2022) calculates the similarity between multiple views of images to distinguish false negatives and true hard negatives. Chen et al. (2022) leverages the property that similar instances are closer in the representation space to incrementally detect false negatives in vision tasks. Zhou et al. (2022) uses another external model to measure the similarity of two sentence representations and selects pairs whose similarity scores are higher than the threshold as negative pairs. In code representation pre-training, Li et al. (2022c) handles it in an iteratively adversarial manner. However, the false negative problem in the fine-tuning of code search is not investigated yet, and it also suffers due to code duplication in code corpora as mentioned by Lopes et al. (2017); Allamanis (2019). The above-mentioned works can be seen as a special case of the proposed Soft-InfoNCE loss.

## 8 Conclusion

In this work, we revisit the commonly used InfoNCE loss in code search and analyze its drawback during fine-tuning. By simply inserting weight terms, we propose Soft-InfoNCE to model the potential relation of negative codes explicitly. We further theoretically analyze its effect on representation distribution, mutual information estimation, and superiority over other loss functions and false negative cancellation methods. We evaluate our Soft-InfoNCE loss on several datasets and models. Experiment results demonstrate the effectiveness of our approach, and justify the theoretical analysis.

## Acknowledgement

We thank the anonymous reviewers for their helpful comments and suggestions. This research is supported by Alibaba-NTU Singapore Joint Research Institute (JRI), Nanyang Technological University, Singapore.

## Limitations

There are mainly three limitations of this work. First, we propose and evaluate three methods to estimate similarity scores $sim_{ij}$ by using BM25, trained models and SimCSE. Since we mainly focus on the justification of Soft-InfoNCE loss itself in this work, the estimation methods are not fully investigated. Which method is better? Are there any other more efficient and precise estimation methods? We leave this as our future work. Second, in this work, we aim to show and justify the effectiveness of Soft-InfoNCE over InfoNCE. For the efficiency of the two, since there is an additional step to estimate the weight of each negative pair, the training time-cost of Soft-InfoNCE is greater than InfoNCE. We compare them in Table 8 and leave improving the efficiency of Soft-InfoNCE as our future work. Third, as we claimed in the paper, this work focuses on the order of negative codes for a given query. However, the widely adopted benchmarks only consist of binary labels, positive and negative. As a result, we could only demonstrate its effectiveness by indirectly showing that it can help models learn better representations.

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

## A Survey on the adoption of InfoNCE loss

To show the dominant adoption of InfoNCE, we conduct a survey on recent code search studies. We collect the papers on top-tier conferences[3] from 2020 and then filter out researches that focus on model training, as shown in Table 4. As we can see, among 16 papers there are only 3 papers that use other loss functions for training.

| Model | InfoNCE |
|---|---|
| RoBERTa (code) (Feng et al., 2020) | ✓ |
| CodeBERT (Feng et al., 2020) | ✓ |
| GraphCodeBERT (Guo et al., 2021) | ✓ |
| UniXCoder (Guo et al., 2022) | ✓ |
| SyncoBERT (Wang et al., 2021) | ✓ |
| SCodeR (Li et al., 2022c) | ✓ |
| CoCoSoDa (Shi et al., 2022a) | ✓ |
| CodeRetriever (Li et al., 2022b) | ✓ |
| RACS (Li et al., 2022a) | ✓ |
| CrossCS (Shi et al., 2022c) | ✓ |
| Code-MVP (Wang et al., 2022) | ✓ |
| cpt-code (Neelakantan et al., 2022) | ✓ |
| Corder (Bui et al., 2021) | ✓ |
| TranCS (Sun et al., 2022) | ✓ |
| CoCLR (Huang et al., 2021) | ✗ |
| CQIL (Gu et al., 2021) | ✗ |
| GSMM (Shi et al., 2022b) | ✗ |

Table 4: The adoption of InfoNCE loss on the training of recent code search models.

---

[3]Specifically, venues ranked A or B under the section of Artificial Intelligence and Software Engineering in the CCF ranking: `https://ccf.atom.im/`.

## B Proof

### B.1 Proof of Theorem 1

$L_{unif}$

$$= \frac{1}{N} \sum_{i=1}^{N} \left[ \log \left[ \exp(q_i \cdot c_i) \right. \right.$$
$$\left. \left. + \sum_{j \neq i}^{N} w_{ij} \cdot \exp(q_i \cdot c_j) \right] \right] \quad (16)$$

$$\geq \frac{1}{N} \sum_{i=1}^{N} \left[ \log \left[ \sum_{j \neq i}^{N} \frac{\beta - \alpha \cdot sim_{ij}}{\beta - \frac{\alpha}{N-1} \sum_{j \neq i}^{N} sim_{ij}} \right. \right.$$
$$\left. \left. \cdot \exp(q_i \cdot c_j) \right] \right] \quad (17)$$

$$\geq \frac{1}{N} \sum_{i=1}^{N} \left[ \log \left[ \sum_{j \neq i}^{N} \frac{\beta - \alpha \cdot sim_{ij}}{\beta - \frac{\alpha}{N-1} \sum_{j \neq i}^{N} sim_{ij}} \right. \right.$$
$$\left. \left. \cdot \exp(q_i \cdot c_j) \right] \right] - \frac{1}{N} \sum_{i=1}^{N} \log(N-1)$$

$$= \frac{1}{N} \sum_{i=1}^{N} \left[ \log \sum_{j \neq i}^{N} \frac{\beta - \alpha \cdot sim_{ij}}{\beta N - \alpha - 1} \exp(q_i \cdot c_j) \right]$$

$$\geq \frac{1}{N} \sum_{i=1}^{N} \sum_{j \neq i}^{N} \frac{\beta - \alpha \cdot sim_{ij}}{\beta N - \alpha - 1} \log \exp(q_i \cdot c_j)$$
$$\quad (18)$$

$$\geq \frac{1}{N} \sum_{i=1}^{N} \sum_{j \neq i}^{N} \frac{\beta - \alpha \cdot sim_{ij}}{\beta N - \alpha - 1} \log \exp(q_i \cdot c_j)$$
$$- \frac{1}{N} \sum_{i=1}^{N} \sum_{j \neq i}^{N} \frac{\beta - \alpha \cdot sim_{ij}}{\beta N - \alpha - 1} \log \sum_{j \neq i}^{N} \exp(q_i \cdot c_j)$$
$$\quad (19)$$

$$= \frac{1}{N} \sum_{i=1}^{N} \sum_{j \neq i}^{N} \frac{\beta - \alpha \cdot sim_{ij}}{\beta N - \alpha - 1} \log P_\theta(c_j | q_i)$$

$$= \frac{1}{N(\beta N - \alpha - 1)} \sum_{i=1}^{N} \sum_{j \neq i}^{N} (\beta - \alpha sim_{ij})$$
$$\cdot \log P_\theta(c_j | q_i) \quad (20)$$

$$= \frac{1}{N(\beta N - \alpha - 1)} \sum_{i=1}^{N} \left[ \beta \sum_{j \neq i}^{N} \log P_\theta(c_j | q_i) \right.$$
$$\left. + \alpha KL(\mathbf{S_i} | P_\theta(c_j | q_i)) + \alpha \sum_{j \neq i}^{N} sim_{ij} \log sim_{ij} \right]$$

We drive from Eq.(16) to Eq.(17) by ignoring the result of positive pair that is always

greater than 0. By subtracting a constant term $\frac{1}{N} \sum_{i=1}^{N} \log(N-1)$ and applying Jensen's Inequality, we get Eq.(18). Then, we subtract Eq.(18) from $\frac{1}{N} \sum_{i=1}^{N} \sum_{j \neq i}^{N} \frac{\beta - \alpha \cdot sim_{ij}}{\beta N - \alpha - 1} \log \sum_{j \neq i}^{N} \exp(q_i \cdot c_j)$. According to previous studies (Li et al., 2020; Zhou et al., 2022), transformer-based models learn an anisotropic embedding space which means that nearly any dot product of two representations is greater than 0. We also find similar phenomena in the context of code search models. Thus, the inequality in Eq.(19) holds in most cases. To further satisfy the condition, we clamp the dot product results that are less than 0 to be 0. Finally, by turning the subtraction outside the logarithms into division inside it, we get the probability $P_\theta(c_j | q_i)$ that refers to the similarity of $q_i$ and $c_j$. Therefore, we reach the conclusion in Theorem 1. And we could find that the constant term is:

$$\frac{1}{N(\beta N - \alpha - 1)} \sum_{i=1}^{N} \left[ \alpha \sum_{j \neq i}^{N} sim_{ij} \log sim_{ij} \right]$$

which is related to $\mathbf{S_i} = \{sim_{ij}\}_{j \neq i}^{N}$.

### B.2 Proof of Theorem 2

The proof only discusses the effect of BCE loss for negative pairs. For the optimization of Positive pairs, BCE loss is equal to Soft-InfoNCE loss.

$L_{BCE}$

$$= -\frac{1}{N^2} \sum_{i=1}^{N} \sum_{j=1}^{N} \left[ sim_{ij} \cdot \log \frac{\exp(q_i \cdot c_j)}{\sum_{j \neq i}^{N} \exp(q_i \cdot c_j)} \right.$$
$$\left. + (1 - sim_{ij}) \cdot \log \left( 1 - \frac{\exp(q_i \cdot c_j)}{\sum_{j \neq i}^{N} \exp(q_i \cdot c_j)} \right) \right]$$

$$= -\frac{1}{N^2} \sum_{i=1}^{N} \sum_{j=1}^{N} \left[ \log(1 - P_\theta(c_j | q_i)) \right.$$
$$\left. + sim_{ij} \cdot \log \frac{\exp(q_i \cdot c_j)}{-\exp(q_i \cdot c_j) + \sum_{j \neq i}^{N} \exp(q_i \cdot c_j)} \right]$$
$$\quad (21)$$

$$\leq -\frac{1}{N^2} \sum_{i=1}^{N} \sum_{j=1}^{N} \left[ \log(1 - P_\theta(c_j | q_i)) \right.$$
$$\left. + sim_{ij} \cdot \log(P_\theta(c_j | q_i)) \right] \quad (22)$$

$$= \frac{1}{N^2} \sum_{i=1}^{N} \left[ -\sum_{j=1}^{N} \log(1 - P_\theta(c_j | q_i)) \right.$$

| Model | Ruby | Python | Java | JavaScript | PHP | Go | Overall |
|---|---|---|---|---|---|---|---|
| CodeBERT | | | | | | | |
| -BCE | 0.603 | 0.615 | 0.639 | 0.555 | 0.572 | 0.863 | 0.641 |
| -Weighted InfoNCE | 0.608 | 0.609 | 0.634 | 0.553 | 0.566 | 0.861 | 0.639 |
| -KL Regularization | 0.615 | 0.623 | 0.643 | 0.557 | 0.586 | 0.861 | 0.648 |
| -Soft-InfoNCE | **0.666** | **0.668** | **0.670** | **0.607** | **0.623** | **0.887** | **0.687** |
| GraphCodeBERT | | | | | | | |
| -BCE | 0.705 | 0.648 | 0.669 | 0.571 | 0.597 | 0.883 | 0.679 |
| -Weighted InfoNCE | 0.695 | 0.633 | 0.655 | 0.608 | 0.588 | 0.865 | 0.674 |
| -KL Regularization | 0.691 | 0.658 | 0.677 | 0.620 | 0.611 | 0.869 | 0.688 |
| -Soft-InfoNCE | **0.721** | **0.700** | **0.702** | **0.656** | **0.657** | **0.894** | **0.722** |
| UniXCoder | | | | | | | |
| -BCE | 0.721 | 0.666 | 0.669 | 0.571 | 0.608 | 0.883 | 0.686 |
| -Weighted InfoNCE | 0.729 | 0.689 | 0.697 | 0.649 | 0.623 | 0.894 | 0.714 |
| -KL Regularization | 0.729 | 0.692 | 0.699 | 0.659 | 0.637 | 0.882 | 0.716 |
| -Soft-InfoNCE | **0.753** | **0.726** | **0.731** | **0.699** | **0.684** | **0.913** | **0.751** |

Table 5: Results of different loss functions on six programming languages under MRR.

$$+ KL(\mathbf{S_i}|P_\theta(c_j|q_i)) + \sum_{j \neq i}^{N} sim_{ij} \log sim_{ij} \Bigg]$$

By removing the $- \exp(q_i \cdot c_j)$ in the denominator of the second term in Eq.(21), the inequality to Eq.(22) holds. Therefore, we reach the conclusion in Theorem 2. And we could find that the constant term is:

$$\frac{1}{N^2} \sum_{i=1}^{N} \sum_{j \neq i}^{N} sim_{ij} \log sim_{ij}$$

which is related to $\mathbf{S_i} = \{sim_{ij}\}_{j \neq i}^{N}$.

## B.3 Proof of Proposition 1

$$
\begin{aligned}
& L_W - L_{Soft} \\
& = - \sum_{j=1}^{N} sim_{ij} \cdot \log \frac{\exp(q_i \cdot c_j)}{\sum_{j=1}^{N} \exp(q_i \cdot c_j)} \\
& \quad + \log \frac{\exp(q_i \cdot c_i)}{\sum_{j=1}^{N} w_{ij} \exp(q_i \cdot c_j)} \\
& = - \sum_{j=1}^{N} sim_{ij} \cdot \log \frac{\exp(q_i \cdot c_j)}{\sum_{j=1}^{N} \exp(q_i \cdot c_j)} \\
& \quad - \log \Bigg[ \exp(q_i \cdot c_i) + \sum_{j \neq i}^{N} w_{ij} \cdot \exp(q_i \cdot c_j) \Bigg] \\
& \quad + \log \exp(q_i \cdot c_i)
\end{aligned}
\tag{23}
$$

$$
\begin{aligned}
& \leq - \sum_{j=1}^{N} sim_{ij} \cdot \log \frac{\exp(q_i \cdot c_j)}{\sum_{j=1}^{N} \exp(q_i \cdot c_j)} \\
& \quad - \sum_{j \neq i}^{N} w_{ij} \cdot \log \exp(q_i \cdot c_j) \\
& = - \sum_{j=1}^{N} sim_{ij} \cdot \log \exp(q_i \cdot c_j) \\
& \quad + 2 \log \sum_{j=1}^{N} \exp(q_i \cdot c_j) \\
& \quad - \sum_{j \neq i}^{N} \frac{(N-1)(1 - sim_{ij})}{N-2} \log \exp(q_i \cdot c_j) \\
& = - \sum_{j \neq i}^{N} \frac{N - 1 - sim_{ij}}{N - 2} \log \exp(q_i \cdot c_j) \\
& \quad + 2 \log \sum_{j=1}^{N} \exp(q_i \cdot c_j) - \log \exp(q_i \cdot c_i) \\
& = - \sum_{j \neq i}^{N} \frac{N - 1 - sim_{ij}}{N - 2} \log P_\theta(c_j|q_i) \\
& \quad - \log P_\theta(c_i|q_i)
\end{aligned}
\tag{24}
$$

We derive Eq.(24) from Eq.(23) by first ignoring $\exp(q_i \cdot c_i)$ within the logarithm and then applying Jensen's Inequality. Therefore, we get:

$$L_{Soft} \geq L_W + \log P_\theta(c_i|q_i)$$
$$+ \sum_{i \neq j} \frac{N - 1 - sim_{ij}}{N - 2} \log P_\theta(c_j|q_i)$$

The equilibrium satisfies when all code fragments in the batch are actually positives and the model perfectly predicts them.

## C Experiment Settings

### C.1 Dataset Statistics

The dataset statistics of CodeSearchNet are shown in Table 6.

| Language | Training | Validation | Test | Codebase |
|---|---|---|---|---|
| Ruby | 24,927 | 1,400 | 1,261 | 4,360 |
| Python | 251,820 | 13,914 | 14,918 | 43,827 |
| Java | 164,923 | 5,183 | 10,955 | 40,347 |
| JavaScript | 58,025 | 3,885 | 3,291 | 13,981 |
| PHP | 241,241 | 12,982 | 14,014 | 52,660 |
| Go | 167,288 | 7,325 | 8,122 | 28,120 |

Table 6: CodeSearchNet dataset statistics.

### C.2 Hyper-parameter Settings

The settings of $\alpha$ and $\beta$ are shown in Table 7. The settings of $\alpha$ and $\beta$ in BM25 are different because we calculate BM25 scores only based on in-batch data, which results in similar scores for negative pairs. Thus, to better distinguish negative pairs, we set higher $\alpha$ compared with the other two estimation methods. And $t$ is used to tune the distribution of negative similarity scores $sim_{ij}$ similar to softmax distribution, which is in accordance with our hypothesis. Note that all the calculated weights $w_{ij}$ are clamped to be greater than 0.1.

| Estimation Methods | $\alpha$ | $\beta$ | $t$ |
|---|---|---|---|
| BM25 | 1.5 | 0.5 | 1.0 |
| SimCSE | 1.3 | 0.7 | 0.1 |
| Trained Model | 1.3 | 0.7 | 5.0 |

Table 7: $\alpha$, $\beta$ and $t$ of different estimation methods.

| Loss | Time per batch |
|---|---|
| InfoNCE | 0.75s |
| Soft-InfoNCE | |
| -BM25 | 0.77s |
| -SimCSE | 0.81s |
| -Trained Model | 0.98s |

Table 8: Training time efficiency comparison of InfoNCE and Soft-InfoNCE.

## D Detailed Experimental Results

### D.1 Efficiency Comparison between Soft-InfoNCE and InfoNCE

To compare the time efficiency, we take CodeBERT as an example, and calculate time cost per batch based on the average value of 30 epochs, using the same batch size for training. The results shown in Table 8 reveal that the implementation of Soft-InfoNCE has a negligible increase in time overhead while improving performance. As discussed in Limitations, we aim to show and justify the effectiveness of Soft-InfoNCE over InfoNCE in this work and we leave the efficiency improvement of Soft-InfoNCE as our future work.

### D.2 Case Study

We think that it would be beneficial to see a more detailed analysis. Thus, we analyze the eight examples in the right part of Fig.2 in detail. We first list out all eight examples. The natural language description is written in the caption above each code snippet.

The positive example is the first one and the left seven examples are negative ones.

Listing 1: Extracts encoded genotype data from binary formatted file.

```
def extract_genotypes(self, bytes):
    genotypes = []
    for b in bytes:
        for i in range(0, 4):
            v = ((b>>(i*2)) & 3)
            genotypes.append(self.geno_conversions[
                v])
    return genotypes[0:self.ind_count]
```

Listing 2: Gets the SasLogicalInterconnects API client.

```
def sas_logical_interconnects(self):
    if not self.__sas_logical_interconnects:
        self.__sas_logical_interconnects =
            SasLogicalInterconnects(self.
            __connection)
    return self.__sas_logical_interconnects
```

Listing 3: Read the specified number of bytes from the stream.

```python
def read_bytes(self, length) -> bytes:
    value = self.stream.read(length)
    return value
```

Listing 4: Copy a full directory structure.

```python
def copy_tree(src, dst, symlinks=False, ignore=[]):
    names = os.listdir(src)
    if not os.path.exists(dst):
        os.makedirs(dst)
    errors = []
    for name in names:
        if name in ignore:
            continue
        srcname = os.path.join(src, name)
        dstname = os.path.join(dst, name)
        try:
            if symlinks and os.path.islink(srcname)
                :
                linkto = os.readlink(srcname)
                os.symlink(linkto, dstname)
            elif os.path.isdir(srcname):
                copy_tree(srcname, dstname,
                    symlinks, ignore)
            else:
                copy_file(srcname, dstname)
        except (IOError, os.error) as exc:
            errors.append((srcname, dstname, str(
                exc)))
        except CTError as exc:
            errors.extend(exc.errors)
    if errors:
        raise CTError(errors)
```

Listing 5: A memoizing recursive Fibonacci to exercise RPCs.

```python
def memoizing_fibonacci(n):
    if n <= 1:
        raise ndb.Return(n)
    key = ndb.Key(FibonacciMemo, str(n))
    memo = yield key.get_async(ndb_should_cache=False
        )
    if memo is not None:
        assert memo.arg == n
        logging.info('memo hit: %d -> %d', n, memo.
            value)
        raise ndb.Return(memo.value)
    logging.info('memo fail: %d', n)
    a = yield memoizing_fibonacci(n - 1)
    b = yield memoizing_fibonacci(n - 2)
    ans = a + b
    memo = FibonacciMemo(key=key, arg=n, value=ans)
    logging.info('memo write: %d -> %d', n, memo.
        value)
    yield memo.put_async(ndb_should_cache=False)
    raise ndb.Return(ans)
```

Listing 6: Enrich the SKOS relations according to SKOS semantics, including subproperties of broader and symmetric related properties.

```python
def enrich_relations(rdf, enrich_mappings,
    use_narrower, use_transitive):
    # 1. first enrich mapping relationships (
        because they affect regular ones)

    if enrich_mappings:
        infer.skos_symmetric_mappings(rdf)
        infer.skos_hierarchical_mappings(rdf,
            use_narrower)

    # 2. then enrich regular relationships

    # related <-> related
    infer.skos_related(rdf)

    # broaderGeneric -> broader + inverse
        narrowerGeneric
    for s, o in rdf.subject_objects(SKOSEXT.
        broaderGeneric):
        rdf.add((s, SKOS.broader, o))

    # broaderPartitive -> broader + inverse
        narrowerPartitive
    for s, o in rdf.subject_objects(SKOSEXT.
        broaderPartitive):
        rdf.add((s, SKOS.broader, o))

    infer.skos_hierarchical(rdf, use_narrower)

    # transitive closure: broaderTransitive and
        narrowerTransitive
    if use_transitive:
        infer.skos_transitive(rdf, use_narrower)
    else:
```

```python
        # transitive relationships are not wanted,
            so remove them
        for s, o in rdf.subject_objects(SKOS.
            broaderTransitive):
            rdf.remove((s, SKOS.broaderTransitive,
                o))
        for s, o in rdf.subject_objects(SKOS.
            narrowerTransitive):
            rdf.remove((s, SKOS.narrowerTransitive,
                o))

    infer.skos_topConcept(rdf)
```

Listing 7: Retrieves all running tunnels for a specific user.

```python
def get_tunnels(self):
    method = 'GET'
    endpoint = '/rest/v1/{}/tunnels'.format(self.
        client.sauce_username)
    return self.client.request(method, endpoint)
```

Listing 8: power down the OpenThreadWpan.

```python
def powerDown(self):
    print '%s call powerDown' % self.port
    if self.__sendCommand(WPANCTL_CMD + 'setprop
        Daemon:AutoAssociateAfterReset false')[0]
        != 'Fail':
        time.sleep(0.5)
        if self.__sendCommand(WPANCTL_CMD + 'reset'
            )[0] != 'Fail':
            self.isPowerDown = True
            return True
        else:
            return False
    else:
        return False
```

As we could see, the positive query could be summarized as extracting data from a binary file. For SimCSE and BM25, the estimated weights are calculated based on the natural language descriptions of positive codes and negative codes. The second negative example contains tokens like "read from" and "bytes", which makes it perform like reading data from a file as well. Thus, BM25 and SimCSE consider it as the most similar negative sample hence predicting a small weight. However, the natural language description is deceptive. The code of the second negative example in fact reads bytes not based on a file directory but on a given number. On the contrary, since weights that are predicted by trained models are calculated by the positive query and negative codes, real potential false negative samples are captured like the third negative code. We also find that there are cases where BM25 and SimCSE perform better than trained models when the code snippets are too long and complicated which makes it hard for trained models to capture the main purpose of the code.

### D.3 Comparison with Other Loss Functions

Table 5 shows detailed results on each programming language. Table 2 calculates the average value of different programming languages to demonstrate the overall performance.