# OpenReview forum: "Rethinking Negative Pairs in Code Search"
_EMNLP/2023/Conference — EMNLP 2023 Main_

### Official Review · Reviewer_ftqB · 2023-08-02

**Soundness:** 3

**Excitement:**

4: Strong: This paper deepens the understanding of some phenomenon or lowers the barriers to an existing research direction.

**Paper Topic And Main Contributions:**

This paper introduces Soft-InfoNCE, a weighting mechanism in the contrastive loss (InfoNCE). The key hypothesis of this work is to reduce the negative impact of false negatives by giving low weight scores to (estimated) false negative pairs. The weighting scores are obtained by 1) lexical matching (BM25), 2) unsupervised learning (SimCSE), or 3) supervised learning (a trained model with InfoNCE). The experimental results in the CodeSearchNet benchmark, with 6 different languages, generally show the effectiveness of Soft-InfoNCE over InfoNCE.

**Questions For The Authors:**

* (major) Please mention distinct challenges in finetuning stage, so that the existing approaches are not easily applied.

* Regarding the code duplications in line 56 and 183, it would be more helpful if an analysis on the finetuning corpus used in this study (CodeSearchNet) is supported, rather than only adding citations.

* (minor) I think there are some works with opposite intuitions: adversarial training like [2] and minimizing the effect of false negatives [1]. From the perspective of adversarial training, the performance improvement might be come from learning more (lexical) bias. It could be interesting to evaluate the proposed method on AdvTest, and see whether the method incurs such bias.

* In line 488, what is the previous statement?


[2] Li, X., Guo, D., Gong, Y., Lin, Y., Shen, Y., Qiu, X., Jiang, D., Chen, W., & Duan, N. (2022). Soft-Labeled Contrastive Pre-Training for Function-Level Code Representation. In Findings of the Association for Computational Linguistics: EMNLP 2022 (pp. 118–129). Association for Computational Linguistics.

**Reasons To Accept:**

* The approach is simple but effective

**Reasons To Reject:**

* (major) In line 73-80, authors argue that, despite there exist similar approaches during pre-training, this work is novel as it conducts during finetuning. However, I feel this work is incremental unless there are distinct challenges that occur in finetuning stage, but not in pre-training.


* In Section 4.2, though line 301-309 is the key part, it does not theoratically explain why the incorporating $w_{ij}$ results in more precise estimation. (Section 4 in [1] might be helpful for better presentation.) In the current presentation, the possibility of mispredicting $w_{ij}$ is not addressed, which may cause the sampling distribution far from the real distribution. Also, line 279-299 may fit to lie in Preliminaries, and can be summarized more.

* Figures should be better presented: Figure 1 is not informative, and for Figure 2, it was hard to understand the meaning of y-axis in the left figure (samples sorted by the similarity scores?).  And for line 488-494, is that mean the scores from models are bad because of the ignorance of potential similairty in neg codes, thus other methods who have the contradictory scores (e.g. SimCSE) are better? Authors may need to show the code samples (especially those recieved contradictory scores; e.g., sample 2 of the x-axis in Fig 2's right figure)


[1] Chuang, Ching-Yao, et al. "Debiased contrastive learning." Advances in neural information processing systems 33 (2020): 8765-8775.

**Reproducibility:**

5: Could easily reproduce the results.

**Reviewer Confidence:**

4: Quite sure. I tried to check the important points carefully. It's unlikely, though conceivable, that I missed something that should affect my ratings.

**Typos Grammar Style And Presentation Improvements:**

* Figure 1 is too ambiguous. Please checkout Figure 1 in [1].

* Please update Figure 2, considering the item 3 in `Reasons to Reject'.

* line 56, 183, 295: Please use \citet for textual citation.

* Section 4.2 should be better presented, especially from line 301 to 309. Section 4 in [1] might be helpful for revision.

* line 296: Eq.(8) (Not important, but it would be good to align notations: Fig.X Eq.(X) Sec.X or Figure X Eq (X) Section X)

* line 651: Soft-InfoNCE

* line 550: $\dfrac{\alpha+\beta}{2}=1$? (as $\alpha = 1.3$, $\beta = 0.7$)

---

> ### Author Rebuttal · Authors · 2023-08-28
>
> We are sincerely grateful for the time volunteered to review our work. We are glad you appreciate that our proposed approach is simple and effective.
>
> > Q1. Typos Grammar Style And Presentation Improvements
>
> We thank you for your careful consideration of typos and potential presentation improvements. We will revise them in our next version according to your suggestion.
>
> > Q2. (major) Please mention distinct challenges in finetuning stage, so that the existing approaches are not easily applied.
>
> There are two works in code search that also address the false negative problem during pre-training, CodeRetriever [1] and SCodeR [2]. However, they can not be easily applied to fine-tuning because they require large amounts of data. As we know, during fine-tuning, we often can not have nearly the same size corpus as pre-training. We will discuss this in detail then.
>
> * CodeRetriever: It proposes a data mining approach to mine potential positive pairs from the pre-training corpus.
>
> * SCodeR: It introduces a discriminator and trains the encoder in an adversarial training style to minimize the effect of false negative samples.
>
> To support our statement, we randomly sampled 20000 pairs from the CodeSearchNet Python and Java datasets (around 10% of the original dataset), respectively. And compare our methods with CodeRetriever and SCodeR. We still take the MRR as evaluation metric.
>
> | Methods | Python | Java |
> | ---- | ---- | ---- |
> | Default (InfoNCE) | 0.320 | 0.355|
> | CodeRetriever | 0.322 | 0.354 |
> | SCodeR | 0.298| 0.337 |
> | **Ours (Soft-InfoNCE)** | **0.334** | **0.363** |
>
> As we can see in the Table, our approach could still improve the performance while CodeRetriever and SCodeR are even outperformed by merely training with InfoNCE. We find that there are only 17 and 23 pairs mined from the Python and Java datasets by CodeRetriever, which has little influence on improving the performance. For SCodeR, we think that the small amount of data can not support the training of the discriminator hence deteriorating adversarial learning. As a result, the performance even drops.
>
> **In conclusion, the distinct challenge for fine-tuning is the small corpus size, which makes pre-training methods not easily applied during fine-tuning.**
>
> We agree with you that it is important to mention this distinct challenge to show the necessity of addressing the false negative problem during fine-tuning. We will add this discussion in our next version.
>
> [1] Li X, Gong Y, Shen Y, et al. CodeRetriever: A Large Scale Contrastive Pre-Training Method for Code Search[C]//Proceedings of the 2022 Conference on Empirical Methods in Natural Language Processing. 2022: 2898-2910.
>
> [2] Li X, Guo D, Gong Y, et al. Soft-Labeled Contrastive Pre-training for Function-level Code Representation[J]. arXiv preprint arXiv:2210.09597, 2022.
>
> > Q3. I feel this work is incremental unless there are distinct challenges that occur in finetuning stage, but not in pre-training.
>
> While we have addressed the distinct challenges during fine-tuning in our response to Q2, we do not consider our work to be merely incremental. **Besides proposing Soft-InfoNCE for addressing the problem, another important contribution of our work is the theoretical justification.** We conduct theoretical analysis to show that Soft-InfoNCE loss can control the distribution of learnt representations and reduce the variance in mutual information estimation. We also prove the superiority of Soft-InfoNCE loss over other choices of design.
>
> > Q4. In Section 4.2, though line 301-309 is the key part, it does not theoratically explain why the incorporating $w_{ij}$ results in more precise estimation. (Section 4 in [1] might be helpful for better presentation.)
>
> **As mentioned in Lines 297-299, InfoNCE loss builds a Monte-Carlo estimation sampling from a uniform distribution. And incorporating $w_{ij}$ can be considered as using the importance sampling strategy in Monte-Carlo estimation. The idea behind the importance sampling is to reduce the statistical uncertainty of Monte-Carlo estimation by drawing samples from a given distribution instead of a uniform distribution, which makes it focus on the most important region of the space [3, 4].**
>
> In the context of our paper, we want to estimate the expectation of $\frac{p(c_j|q_i)}{p(c_j)}$ according to a real distribution between $q_i$ and negative codes $c_j$. However, the negative codes in a batch are randomly sampled which follows a uniform distribution. To bridge this gap and get an unbiased estimator, importance sampling is adopted by inserting a weight term $w_{ij}$.
> We denote the uniform distribution as $p$ and the real distribution as $q$. Thus, calculating expectations based on $q$ could be formulated as:
>
> $$
> \mathbf{E_{c_j \sim q}} \left[ \frac{p(c_j|q_i)}{p(c_j)} \right] =\mathbf{E_{c_j \sim p}}  \left[ \frac{p(c_j|q_i)}{p(c_j)} \cdot \frac{q(c_j)}{p(c_j)} \right]
> $$
>
> Here, the uniform distribution $p$ is $\frac{1}{N-1}$, and the constructed real distribution is $\frac{1-sim_{ij}}{N-2}$. Thus, $\frac{q(c_j)}{p(c_j)} = w_{ij}$ when we take both $\alpha$ and $\beta$ as 1 according to Eq.4. In Lines 305-309, we discuss the validity of constructed real distribution $q$ and we empirically verify it in the left part of Fig.2. As we assumed, the real distribution is inversely proportional to $sim_{ij}$ and follow a long-tailed distribution.
>
> **Theoretical analysis in Section 4 of your mentioned paper [5] is a special case of our study.** In that paper, the authors theoretically analyze the bias only based on true negative samples and false negative samples. However, in the context of code search, it is not a discrete case. Some negative samples are more negative while some are less negative, which is a continuous case. In Table 3 of our paper, we also compare our Soft-InfoNCE with merely dividing negative samples into true and false ones. Our Soft-InfoNCE outperforms the discrete case.
>
> We agree with you that since we skipped some steps in the theoretical analysis, this part is a bit unclear. We will add this detailed derivation in our next version.
>
> [3] Arouna, Bouhari (2004). "Adaptative Monte Carlo Method, A Variance Reduction Technique". Monte Carlo Methods and Their Applications.
>
> [4] Monte Carlo Methods – Lecture Notes, N. Shimkin, Spring 2015
>
> [5] Chuang, Ching-Yao, et al. "Debiased contrastive learning." Advances in neural information processing systems 33 (2020): 8765-8775.
>
> > Q5. In the current presentation, the possibility of mispredicting $w_{ij}$ is not addressed, which may cause the sampling distribution far from the real distribution.
>
> As we discussed in the ‘Limitations’ Section, the estimation methods are not fully investigated because in this paper we mainly focus on the justification of Soft-InfoNCE loss itself.
>
> Yes, we admit that there is a possibility of mispredicting $w_{ij}$. As we can see, in the right part of Fig.2, for the same batch of sample, different estimation methods may even predict contradictory weights. There must be correct estimation and wrong estimation. **However, as shown in Table 1, overall Soft-InfoNCE could reach better performance, which indicates that Soft-InfoNCE is robust to mispredicted $w_{ij}$.** The detailed analysis of the Fig.2 can be found in our response to Q7.
> We leave investigations of estimation methods as our future work.
>
> > Q6. In line 488, what is the previous statement? And for line 488-494, is that mean the scores from models are bad because of the ignorance of potential similairty in neg codes, thus other methods who have the contradictory scores (e.g. SimCSE) are better?
>
> We find that there are contradictory predictions when we use InfoNCE-tuned code search models to predict $w_{ij}$. This phenomenon indicates that although existing code search models could find true code snippets well, they cannot recognize potential relevance among negative codes. This is the core motivation for proposing Soft-InfoNCE, which we give an example in Lines 177-182.
> Sorry for not making this part clear, we will rewrite these lines in our next version.
>
> > Q7.  Regarding the code duplications in line 56 and 183, it would be more helpful if an analysis on the finetuning corpus used in this study (CodeSearchNet) is supported, rather than only adding citations. Authors may also need to show the code samples (especially those recieved contradictory scores; e.g., sample 2 of the x-axis in Fig 2's right figure)
>
> Yes, we agree with you that adding case studies will help readers better understand the problems we address, and the successful and failed cases of our proposed approach. **Reviewer swyC also mentions this point. Please see our response to Reviewer swyC (Response to Q2) for a detailed case study if you are interested.** We will add this case study in our next version.
>
> > Q8. (minor) I think there are some works with opposite intuitions: adversarial training like [2] and minimizing the effect of false negatives [1].
>
> The adversarial training method SCodeR is the paper we discuss in the response to Q2. It requires large amounts of data which often can not be satisfied during fine-tuning.
>
> The other paper is discussed in our response to Q4. This paper could be considered as a special case of our proposed method. And our Soft-InfoNCE outperforms it as shown in Table 3.

---

### Official Review · Reviewer_swyC · 2023-08-04

**Soundness:** 4

**Excitement:**

4: Strong: This paper deepens the understanding of some phenomenon or lowers the barriers to an existing research direction.

**Paper Topic And Main Contributions:**

This paper addresses the problem of improving contrastive learning for code search models. It identifies significant issues with the widely used InfoNCE loss function, namely the existence of false negative samples and the inability to differentiate the potential relevance of negative samples. The paper's main contribution is the proposal of a novel Soft-InfoNCE loss function, which includes weight terms to address the identified problems. The authors suggest three methods for estimating the weights of negative pairs and demonstrate theoretically and empirically that their proposed Soft-InfoNCE loss function improves the control of the distribution of learned code representations and provides a more precise mutual information estimation.

**Reasons To Accept:**

The paper's main strength lies in its novel approach to improving contrastive learning for code search models. The Soft-InfoNCE loss function is an innovative solution that addresses significant issues with existing methods. The authors' thorough theoretical analysis and extensive empirical evidence provide strong support for the effectiveness of their proposed method. If presented at the conference, this paper could significantly advance the field of contrastive learning for code search models and inspire further research and development in this area.

**Reasons To Reject:**

While the paper presents a novel and theoretically grounded approach, it does have some weaknesses. The paper could provide more thorough comparisons with other potential methods for addressing the identified issues with the InfoNCE loss function. Additionally, while the authors provide extensive empirical evidence supporting their proposed method, it would be beneficial to see a more detailed analysis of the results, including a discussion of any limitations or potential drawbacks of the Soft-InfoNCE loss function.




**Reproducibility:**

5: Could easily reproduce the results.

**Reviewer Confidence:**

2: Willing to defend my evaluation, but it is fairly likely that I missed some details, didn't understand some central points, or can't be sure about the novelty of the work.

---

> ### Author Rebuttal · Authors · 2023-08-28
>
> Thank you for taking the time to review our work. We are sincerely grateful for your feedback and are glad you appreciated that our work could significantly advance the field and inspire further research in this area.
>
> > Q1. The paper could provide more thorough comparisons with other potential methods for addressing the identified issues with the InfoNCE loss function.
>
> To the best of our knowledge, our work is the first work to address the false negative problem with InfoNCE in fine-tuning code search models. We compare Soft-InfoNCE with Weighted InfoNCE, which is another choice of design, both theoretically and empirically in Proposition 1 and Table 2. As we could see, Soft-InfoNCE upper bounds Weighted InfoNCE and outperforms it on a dataset that consists of six programming languages.
>
> There are also some other ways to address the false negative problem, although not using InfoNCE loss function. We also discuss and compare Soft-InfoNCE with BCE loss and KL divergence regularization and show its superiority in Section 4.3 and Table 2.
>
> At the same time, there are two works addressing the false negative problem during pre-training. However, those approaches could not be applied during fine-tuning since they require large amounts of data. Reviewer ftqB also mentioned this point. Please see our response to Reviewer ftqB (Response to Q2) for a detailed discussion of those two works if you are interested.
>
> > Q2. It would be beneficial to see a more detailed analysis of the results, including a discussion of any limitations or potential drawbacks of the Soft-InfoNCE loss function.
>
> As discussed in the ‘Limitations’ Section, Soft-InfoNCE introduces additional computational cost due to the weight estimation. Besides, it relies on the precision of weight estimation. In this paper, we evaluate three weight estimation methods, and all of them achieve better performance. However, since we aim to show and justify the effectiveness of Soft-InfoNCE in this paper, the potential choices of more precise and efficient estimation methods remain uninvestigated. We leave them as our future work.
>
> We agree that it would be beneficial to see a more detailed analysis. Thus, we analyze the eight examples in the right part of Fig.2 in detail. We first list out all the eight examples:
>
> The positive example is:
> * Extracts encoded genotype data from binary formatted file.
>
> And the seven negative examples are:
> * Gets the SasLogicalInterconnects API client.
> * Read the specified number of bytes from the stream.
> * Copy a full directory structure.
> * A memoizing recursive Fibonacci to exercise RPCs.
> * Enrich the SKOS relations according to SKOS semantics, including subproperties of broader and symmetric related properties.
> * Retrieves all running tunnels for a specific user.
> * power down the OpenThreadWpan
>
> Note that due to the limited space of the rebuttal, we do not show all of the code snippets here, but we will show complete code snippets in the appendix of our next version.
>
> As we could see, the positive query could be summarized as extracting data from a binary file. For SimCSE and BM25, the estimated weights are calculated based on the natural language descriptions of positive codes and negative codes. The second negative example contains tokens like ‘read from’ and ‘bytes’, which makes it perform like reading data from a file as well. Thus, BM25 and SimCSE consider it as the most similar negative sample hence predicting a small weight. However, the natural language description is deceptive. The code of the second negative example is:
>
> ```
> def read_bytes(self, length) -> bytes:
>         value = self.stream.read(length)
>         return value
> ```
>
> This method reads bytes not based on a file directory but on a given number.
>
> On the contrary, since weights that are predicted by trained models are calculated by the positive query and negative codes, real potential false negative samples are captured. The code of the third negative example is:
>
> ```
> def copy_tree(src, dst, symlinks=False, ignore=[]):
>     …
>     if symlinks and os.path.islink(srcname):
>         linkto = os.readlink(srcname)
>         os.symlink(linkto, dstname)
>     …
> ```
>
> However, there are also cases that BM25 and SimCSE perform better than trained models when the code snippets are too long and complicated which makes it hard for trained models to capture the main purpose of the code.
>
> We think this case study will give readers a better understanding of estimation methods. We will add this case study to our next version. However, kindly note that in this paper we do not investigate estimation methods fully but mainly focus on the effectiveness of Soft-InfoNCE, as discussed in the ‘Limitations’ Section.

---

### Official Review · Reviewer_Ls9b · 2023-08-05

**Soundness:** 4

**Excitement:**

4: Strong: This paper deepens the understanding of some phenomenon or lowers the barriers to an existing research direction.

**Paper Topic And Main Contributions:**

This paper discusses contrastive learning, focusing on the role of negative pairs and their varying contributions to the learning process in the context of code search tasks. The paper challenges the conventional contrastive learning assumption that treats all negative pairs equally, highlighting the potential for improved performance when considering the "hardness" level of different negative pairs.
The paper provides an in-depth theoretical understanding of the role of negative pairs in contrastive learning. It introduces Soft-InfoNCE, a new method that assigns weights to negative pairs based on their hardness levels. The paper validates the effectiveness of Soft-InfoNCE through empirical evaluations on code search tasks. The authors discuss the potential for their method to be applied to other contrastive learning tasks and propose future research directions.

**Questions For The Authors:**

A The paper focuses on code search tasks. Could you discuss the potential challenges or considerations when applying Soft-InfoNCE to other NLP tasks?

B The paper does not discuss the computational efficiency of Soft-InfoNCE. Could you provide information on the computational cost of the proposed method compared to other methods, especially considering the additional computations for weight assignment?

C Could you provide more details on the choice of hyperparameters β and γ? It is mentioned that "performance of Soft-InfoNCE is relatively stable with different settings and α = 1.3, β = 0.7 reaches its best performance". Could you give reasons for that?

**Reasons To Accept:**

1 The paper presents a novel theoretical perspective on the role of negative pairs in contrastive learning. This not only challenges the existing assumptions in the field but also provides new insights that can stimulate further research.

2 The proposed Soft-InfoNCE method is also a significant contribution. By assigning different weights to negative pairs based on their hardness levels, the method addresses a key challenge in contrastive learning. The approach is novel and could inspire similar methodologies in other areas of NLP.

3 The authors further provide a comprehensive empirical evaluation of their proposed method. The experiments are well-designed and conducted on a publicly available, multi-language code search dataset, strengthening the validity of the results.

**Reasons To Reject:**

1 While the paper provides valuable insights for contrastive learning in code search tasks, it does not thoroughly explore the implications of their proposed method for other NLP tasks. This somewhat limits the generalizability of the results.

2 The paper does not discuss the computational efficiency of the proposed method. As the Soft-InfoNCE method involves additional computations for weight assignment, it would be important to understand the trade-off between improved performance and increased computational cost.

3 While the authors present the results of their experiments, they do not provide an in-depth analysis of these results. More detailed analysis, including a discussion of cases where the proposed method performs exceptionally well or poorly, could have added depth to the paper.

**Reproducibility:**

4: Could mostly reproduce the results, but there may be some variation because of sample variance or minor variations in their interpretation of the protocol or method.

**Reviewer Confidence:**

4: Quite sure. I tried to check the important points carefully. It's unlikely, though conceivable, that I missed something that should affect my ratings.

---

> ### Author Rebuttal · Authors · 2023-08-28
>
> Thanks a lot for your reviews! We are very encouraged that you found our paper could stimulate further research.
>
> > Q1. The paper focuses on code search tasks. Could you discuss the potential challenges or considerations when applying Soft-InfoNCE to other NLP tasks?
>
> In this paper, we focus on the code search task because there are published papers that discuss the existence of false negative samples and their negative influence on representation learning, as written in Lines 56-60. Thus, we aim to solve the problem of false negative samples in code search.
>
> Intuitively, other NLP tasks may also suffer from the same problem. For example, in passage retrieval, texts that describe Shiba are expected to be retrieved before texts of Rabbit for a given query asking about Corgi because both Corgi and Shiba are dogs. When applying Soft-InfoNCE to passage retrieval, we need to first make sure that the existence of false negatives definitely deteriorates the representation learning. Then, the potential relevance between queries and passages could be captured to a certain extent, although not necessarily extremely accurately. We leave the analysis of the false negative problem and application of Soft-InfoNCE in passage retrieval as our future work.
>
> > Q2. The paper does not discuss the computational efficiency of Soft-InfoNCE. Could you provide information on the computational cost of the proposed method compared to other methods, especially considering the additional computations for weight assignment?
>
> We discuss the computational efficiency of Soft-InfoNCE in Appendix D.1. We record the average time cost per batch during training in Table 8. However, as discussed in ‘Limitations’, in this paper we aim to show and justify the effectiveness of Soft-InfoNCE and we leave the efficiency improvement of it as our future work.
>
> > Q3. Could you provide more details on the choice of hyperparameters β and γ? It is mentioned that "performance of Soft-InfoNCE is relatively stable with different settings and α = 1.3, β = 0.7 reaches its best performance". Could you give reasons for that?
>
> As shown in the left part of Fig.3, we try different α and β choices and find that α=1.3, β=0.7 achieves the best performance. From the right part of Fig.3, we could find that increasing α (decreasing β correspondingly) could make the weights more distinct, which we think could help models better distinguish those negative samples. However, setting α too high also brings risk. That is, the weights of some negative samples are too small. Although those small-weight negatives are less negative compared with other negative samples, models still need to distinguish them with the positive code snippet.
>
> We may also explain from another perspective. As proven in Theorem 1, α controls the weights of the KL divergence term that fits predicted negative similarity to the given similarity scores $S_i$ while β for lowering the predicted negative similarity. Those two terms are contradictory and α=1.3, β=0.7 achieves a balance between the two.
> We will add this discussion in the next version.

---

### Meta-Review · Area_Chair_715u · 2023-09-22

**Recommendation:** 5

**Metareview:**

All the reviewers have agreed that it is interesting to see such work in code search that tackles with false negatives.

The reviewers have raised a few questions, and during the rebuttal period, the authors have largely addressed these questions.

I suggest the authors further follow the suggestions of the reviewers to make this paper more promising, e.g., applications to other NLP fields.

---

### Decision · Program_Chairs · 2023-10-07

**Decision:**

Accept-Main

**Comment:**

All the reviewers have agreed that it is interesting to see such work in code search that tackles with false negatives.

The reviewers have raised a few questions, and during the rebuttal period, the authors have largely addressed these questions.

I suggest the authors further follow the suggestions of the reviewers to make this paper more promising, e.g., applications to other NLP fields.